# The Effects of Imagery Practice on Athletes’ Performance: A Multilevel Meta-Analysis with Systematic Review

**DOI:** 10.3390/bs15050685

**Published:** 2025-05-16

**Authors:** Yiran Liu, Shiao Zhao, Xuda Zhang, Xiaoyu Zhang, Taihe Liang, Ziheng Ning

**Affiliations:** Faculty of Health Sciences and Sports, Macao Polytechnic University, R. de Luís Gonzaga Gomes, Macao 999078, China; p2417052@mpu.edu.mo (Y.L.); p2417093@mpu.edu.mo (X.Z.); p2417184@mpu.edu.mo (X.Z.); tigerleung4177@gmail.com (T.L.); zhning@mpu.edu.mo (Z.N.)

**Keywords:** athletic performance, imagery, athletes, psychological skills training, sports performance

## Abstract

Imagery, a classic technique in psychological training, is gaining momentum in competitive sports. Despite the increasing use of imagery, its effectiveness remains debated. Robust, data-driven conclusions are still lacking. This study seeks to investigate the effects of imagery practice on enhancing athletic performance and determine the ideal dosage of such practice through a systematic review and multilevel meta-analysis. A comprehensive search across seven databases, including SportDiscus, PubMed, PsycINFO, Web of Science, MEDLINE, MEDLINE Complete, and CINAHL, yielded 23,027 studies. These were initially reviewed for title and abstract using ASReview, followed by full-text screening with Covidence. A total of 86 studies with 3593 athletes (2104 males and 1110 females) were included in this meta-analysis. Our findings indicate that imagery practice enhances athletic performance, encompassing agility, muscle strength, tennis and soccer performance, and is applicable to both tennis and soccer athletes. The efficacy of integrating imagery practice with one or two additional psychological skills trainings (PSTs) surpasses that of imagery practice in isolation. Moderation analysis revealed that engaging in imagery practice for approximately ten minutes, three times weekly over a span of one hundred days, produces the strongest performance gains. This review offers recommendations for athletes regarding the implementation of imagery practice in routine training or prior to competitions, thereby providing empirical evidence to optimize psychological training programs in competitive settings.

## 1. Introduction

In the context of elite sports, where marginal performance gains are decisive, psychological skills training (PST) has emerged as a crucial complement to physical conditioning ([75]; [95]). The distinction between victory and defeat is increasingly tenuous, with even minor errors potentially resulting in sportsmen forfeiting the championship. During the 2008 Beijing Olympic Games, the average margin between the first and the fourth places in the men’s rowing events was 1.34%, while for women, it was only 1.03% ([11]). Consequently, among the various PST techniques, mental imagery is frequently adopted by athletes ([102]). However, despite its popularity, systematic and conclusive evidence of its efficacy remains limited, particularly concerning dosage effects and sport-specific applicability.

Mental imagery denotes a multimodal cognitive simulation process that allows for the representation of perceptual information in the mind without actual sensory input ([66]; [95]), characterized as a sensory experience that emulates real-life sight, taste, and sound ([4]). A systematic theoretical framework underpins imagery practice to support its principles. The bio-informational theory posits that narrative imagery elicited by verbal text can activate the brain’s associative memory network, prompting individuals to produce corresponding real reaction information, including autonomic or somatic nervous responses ([49]). Moreover, Carpenter posited that vivid imagery enhances muscle activity and resembles genuine muscle activation, hence facilitating actual motor performance, so this theory is referred to as the psychoneuromuscular theory ([67]). These theoretical foundations justify the investigation of imagery’s physiological and performance outcomes, which we empirically test via meta-analytic synthesis. However, despite this theoretical backing, the actual effectiveness of imagery practice remains uncertain.

Certain experts assert that employing imagery practice ([64]) or integrating it with physical exercise can augment the efficacy of enhancing athletic performance ([53]). Additionally, [60] ([60]) indicated that imagery, self-talk, and goal setting were most advantageous for enhancing athletes’ endurance performance, and imagery practice appears to positively influence the enhancement of motor skill performance, including basketball performance ([14]; [28]), gymnastics performance ([8]; [47]), tennis performance ([50]; [68]), soccer performance ([3]; [83]), among others. Moreover, a meta-analysis determined that imagery practice is an excellent method for improving athletic performance; however, this meta-analysis lacks effective exploration of the dose of imagery and includes relatively few articles, requiring further investigation ([89]). Thus, while the existing evidence suggests that imagery practice holds promise, the methodological constraints and limited scope of past studies highlight the need for more rigorous and comprehensive investigations.

Notwithstanding the increasing data endorsing imagery practice, considerable obstacles persist in its execution. Diverse studies present varying viewpoints on the efficacy of imagery practice in enhancing athletic performance. Recent research asserts that imagery practice, mindfulness training, and PSTP constitute the most effective psychological skills training for enhancing athletes’ performance ([75]). Nevertheless, upon the exclusion of low-quality studies, including subjective performance outcomes and non-randomized controlled trials, the actual effect of PST diminishes significantly ([75]). Furthermore, research indicates that imagery practice based on the physical, environment, task, timing, learning, emotion, and perspective (PETTLEP) model does not enhance the athletic performance of youth soccer players ([74]). Simultaneously, the impact of imagery practice on athletes’ jumping performance is similarly negligible ([6]). Additionally, a meta-analysis revealed that the combination of imagery practice and physical exercise is less helpful in enhancing muscle strength than physical exercise alone ([70]). [90] ([90]) suggested that imagery practice appears to provide advantageous effects solely for the athletic performance of elite athletes.

On the other hand, the ideal dosage of imagery practice for athletes remains largely unexamined by experts. Recent studies indicate that imagery practice for four weeks, lasting fifteen minutes per session, three times weekly, appears to yield the most beneficial results for healthy individuals; however, there is insufficient investigation into the ideal dosage of imagery practice for athletes ([44]; [70]). Therefore, a considerable number of studies have also expressed skepticism regarding the effectiveness of imagery practice in enhancing athletic performance, and few studies have systematically evaluated the optimal dosage of imagery practice. A complete study with a large sample size is needed to reach a strong conclusion on the effectiveness of imagery practice.

This study employed a Bayesian multilevel meta-analysis to assess the impact of imagery practice on athletes’ motor areas. The primary goal was to determine whether imagery enhances athletic performance. If so, we further explored which performance types benefit the most. We also examined whether combining imagery with other psychological skills training yields greater effects than imagery alone, and whether its effectiveness varies by age, gender, or training dosage. Drawing from current yet constrained studies, we suggest two hypotheses: (1) imagery practice has a positive effect on the motor areas; and (2) the effectiveness of imagery practice increases with longer intervention duration.

## 2. Method

This research was registered with PROSPERO (CRD420251001510) and complied with the PRISMA reporting guideline ([38]). A Bayesian meta-analysis was conducted following a systematic review utilizing Covidence, Python version 3, GRADEprofiler, R version 4.4.3, and GetData Graph Digitizer.

### 2.1. Eligibility Criteria

Studies were eligible if they met the following criteria based on the PICOS framework. The participants must have been athletes. Athletes of all ages, irrespective of their health status, were included. Studies must be randomized controlled trials including imagery practice as a treatment. The comparator can be either the absence of psychological training (no practice) or other psychological skills training, excluding imagery practice. Outcomes derived from studies must pertain to athletic performance. The language must be English.

Unpublished articles, as well as master’s and doctoral theses, were excluded. Studies without accessible data for extraction and non-original studies, such as letters, reviews, or editorials, were excluded from the selection procedure. Studies employing various modalities of the same intervention without a control group were excluded, such as different forms of imagery intervention, including textual or video imagery.

### 2.2. Information Sources

A thorough search strategy was formulated employing Medical Subject Headings (MeSH) and free-text search phrases to comprehensively examine seven English databases, including SportDiscus, PubMed, PsycINFO, Web of Science, MEDLINE, MEDLINE Complete, and CINAHL, on 22 February 2025. The keywords and subject headings were conclusively determined by deliberation among the three authors (S.Z., Z.N., and Y.L.). Comprehensive search strings for each database are included in Appendix A. A total of 23,027 studies were retrieved utilizing the Covidence online tool and ASReview in Python 3.13.3 ([96]) for the systematic screening.

### 2.3. Study Selection and Data Collection

All abstracts were evaluated with the assistance of the machine learning technology ASReview to guarantee a rigorous, reproducible procedure ([96]). By training the tool with pertinent and impertinent abstracts, it autonomously forecasts study relevance. ASReview perpetually re-evaluates the complete list of remaining titles or abstracts based on their probability of inclusion. This method diminishes the quantity of abstracts necessitating evaluation by the human reviewer to encompass all eligible papers, hence enhancing the efficiency of the selection process ([43]; [100]). Using a conservative application of the SAFE guideline, inclusion of studies through ASReview was continued until 200 consecutive studies were deemed ineligible ([15]). In the next phase, two reviewers (S.Z. and Y.L.) used the Covidence online tool recommended by PRISMA to screen full texts ([38]). Eligible full-texts were assessed using Extraction 1.0 forms, with discrepancies addressed by the third reviewer ([97]).

### 2.4. Data Items

For each study, the characteristics extracted included authors, publication years, country, intervention, study design, session of imagery practice, sample size, gender, types of athletes, age, training years, and outcomes. The outcomes of motor areas encompassed volleyball performance, tennis performance, swimming performance, subjective performance, sprint, soccer performance, skiing performance, shot performance, rugby performance, racquetball performance, muscle strength, karate performance, jump performance, gymnastics performance, golf performance, field hockey performance, equestrian performance, dance performance, cricket performance, boxing performance, basketball performance, baseball performance, badminton performance, and archery performance.

Data were extracted independently by two authors (S.Z. and Y.L.) using Covidence, with conflicts resolved through discussion with the third author (Z.N.). Data were presented as the means ± standard deviations (M ± SD). We used an online toll called Meta Accelerator to convert data that were not initially in the M ± SD format ([1]). When data were not presented as exact numbers, Get Data Graph Digitizer ([31]) was used to extract data from graphs.

### 2.5. Risk of Bias Assessment

The risk of bias for all the included studies was evaluated independently according to the standards specified in the *Cochrane Handbook for Systematic Reviews of Interventions* ([42]). Two authors (S.Z. and Y.L.) assessed the included studies through the Cochrane risk of bias (ROB2) criteria in RCTs within Covidence. Seven areas of bias were evaluated: (1) random sequence generation; (2) allocation concealment; (3) blinding of the participants and personnel; (4) blinding of the outcome assessment; (5) incomplete outcome data; (6) selective reporting; and (7) other bias. The risk of bias was classified as low, unclear, or high. Following separate evaluations, the writers achieved consensus through deliberation. The final data were documented in an Excel template and subsequently entered into R software 4.4.3 to generate risk-of-bias summary graphs utilizing the robvis package ([61]). Studies exhibiting more than two but fewer than four areas designated as unclear-risk were categorized as having moderate overall risk.

Furthermore, funnel plots were created utilizing the PublicationBias package ([16]) to identify publication bias within the included studies. The plots illustrate both affirmative and non-affirmative studies as round dots positioned on either side of the funnel’s central axis, with a black diamond denoting the total effect size. The grey diamond represents the overall estimate obtained from the non-affirmative research. By configuring the favor_positive parameter to TRUE, the study concentrated explicitly on the bias towards positive outcomes, facilitating a more lucid comprehension of any potential bias in the literature favoring noteworthy discoveries. Evaluating the symmetry of the data points aids in identifying the existence of publication bias. The Egger test was performed via the rma.mv function inside the metafor package. The *p*-value less than 0.05 indicates the presence of potential bias risk. The S-value, derived from the PublicationBias package, indicates the strength of publication bias required to nullify the meta-analytic effect ([58]).

### 2.6. Certainty in Evidence

The quality of evidence in the results of athletic performance was assessed using the GRADE approach, which evaluates the risk of bias, inconsistency, indirectness, and imprecision of effect estimates ([7]). The GRADE approach classifies the quality of evidence as high, moderate, low, or very low.

### 2.7. Statistical Analysis

Bayesian mixed-effects models implemented in the brms package ([18]) in R 4.4.3 were used to analyze variation in effect sizes. We fitted models supposing a normal distribution and incorporated random effects for within-study ID and between-study ID to address within-study and between-study heterogeneity in each outcome based on two formulas:(1)$$\begin{array}{l} I_{within}^{2}=\frac{\tau_{withID}^{2}}{total\_\mathrm{var}iance}\times 100\\ I_{between}^{2}=\frac{\tau_{betweenID}^{2}}{total\_\mathrm{var}iance}\times 100 \end{array}$$

Weakly informative Cauchy priors (mean = 0, SD = 1) were employed for the random effects. The athletic performances were analyzed and categorized based on the types of imagery strategies (imagery alone or imagery practice paired with other psychological skills training), categories of athletes, and athletic performance outcomes. Four models were applied to each set of athletic performance data:(1)Null model to estimate the overall effect sizes. We fitted models on the full dataset for athletic performance.(2)Imagery model. We categorized imagery practice according to its distinct characteristics. The first type included imagery practice that was not integrated with other psychological skills training, whereas the second type consisted of psychological skills training packages (PSTPs) that incorporated imagery practice. Secondly, a comparison was made between these two forms of imagery practice and other psychological skills training, or the absence of psychological training (no practice).(3)Performance model: we categorized the derived athletic performance results according to their characteristics.(4)Athlete model: athletes were categorized according to the type of sport they participated in.

Each model was estimated using four Markov chains, each running for 10,000 iterations. To evaluate the convergence of the Markov chains, we relied solely on the Rhat statistic. Rhat assesses the ratio of between-chain to within-chain variance, where values close to 1 suggest satisfactory convergence ([18]). This metric provides a reliable assessment of convergence, ensuring that chains with Rhat values near 1 have thoroughly explored the posterior distribution, thereby enhancing the reliability of model inference. Bayes factors (BFs) ([56]) were calculated using the reciprocal of the Savage–Dickey density ratio, implemented through the bayesfactor_parameters function in the bayestestR package. BF > 10 indicates strong evidence in favor of the effect ([46]). For complex hierarchical models, the 95% high-density interval (HDI) ([48]) offers a direct and assumption-free representation of the most credible posterior values. This stands in contrast to *p*-values and confidence intervals, which rely on additional assumptions.

### 2.8. Moderation Analysis

In this study, moderation analysis was performed using the brms package in R, incorporating eight moderator variables: age, gender, training experience (in years), disease status, competitive level, sport type (team or individual), PST experience, and the number of PST in PSTPs (PSTP types). PSTP types refer to the number of psychological skills trainings integrated alongside imagery practice within a training package, categorized as combinations of two, three, or four techniques in total (i.e., imagery plus one, two, or three additional psychological strategies). PST experience was defined as whether the athlete had prior exposure to psychological skills training interventions. Three moderators, including imagery practice duration (in days), frequency (sessions per week), and intensity (minutes per session), were examined to determine the optimal dosage of imagery practice for enhancing athletic performance. The analysis was conducted within a Bayesian framework, enabling the estimation of effect sizes while accounting for measurement error and the hierarchical structure of the data.

Furthermore, to evaluate the influence of various moderators on model performance, we initially created a null model devoid of any moderators and subsequently developed distinct models integrating individual moderators. Bayesian R-squared values were calculated for each model using the bayes_R2 function in the brms package to measure the proportion of variance elucidated. To assess the explanatory capacity of models with and without moderators, we illustrated the density distributions of R-squared values. This method offers a clear depiction of the impact of incorporating moderators on model fit, facilitating a more straightforward comparison between the null model and the models that include certain moderators.

## 3. Results

The results include six parts, namely, study selection, characteristics of the included studies, quality assessment, meta-analysis with four models, moderation analysis, quality grade, and publication bias.

### 3.1. Study Selection

The flow chart can be found in Figure 1. From seven databases, 23,027 articles were retrieved and imported into Endnote software for filtering out duplicates; 7931 studies were marked as duplicates and removed automatically by Endnote. The remaining 15,096 articles were imported into the ASReview tool for initial screening, including screening titles and abstracts. With the assistance of machine learning models, a total of 1524 articles were screened, and 234 articles were included in the full-text screening. After full-text screening, 157 articles were excluded due to various reasons, and 77 articles were included in this meta-analysis. Through the citation searching method, an additional 9 articles were included. Ultimately, a total of 86 articles were included in the meta-analysis.

### 3.2. Characteristics of the Included Studies

The detailed characteristics of each included study can be found in Appendix A. We included 86 studies and 3593 athletes, comprising 1110 females and 2104 males, with 379 athletes lacking gender information ([9]; [14]; [17]; [45]; [50]; [51]; [71]; [82], [83], [78]; [84]). Except for seven studies designed as randomized crossover studies ([25]; [37]; [86], [87]; [90]; [94]; [101]), all other studies were randomized parallel-group trials. In 24 studies, a combination of imagery practice and other psychological training was used as intervention methods (28%). Thirty-four studies were from Europe (40%), 17 studies were from America (20%), 16 studies were from Asia (19%), 6 studies were from Africa (7%), and 5 studies were from Oceania (6%). Ten studies did not provide age information ([17]; [20]; [34]; [40]; [59]; [72]; [73]; [92]; [93]; [99]).

### 3.3. Quality Assessment

The risk-of-bias summary can be found in Figure 2, and the specific risk-of-bias graph for each study can be found in Appendix A. Due to the wrong randomized method, such as stratified randomization, some studies were marked as high-risk (6%) ([12]; [19]; [39]; [54]; [57]). More than 80% of the studies did not mention the use of blinding and were therefore rated as unclear-risk. Due to incomplete data, two articles were rated as high-risk (2%) and two articles were rated as medium-risk (2%). Overall, over half of the literature was rated as unclear-risk, thus weakening the robustness of the data results.

### 3.4. Meta Analysis

The meta-analysis was divided into four sections. The Section 1 included the null model, followed by the imagery model, the performance model, and the athlete model. The detailed results in each model can be found in Appendix A. The convergence of the Markov chain across all four models was effectively assessed using the Rhat parameter value in the results. The Rhat value in all the outcomes was approximately 1.0. Consequently, we did not present the results of Markov chain convergence in the article.

### 3.5. Null Model

In the null model, the overall effect size was calculated for athletic performance. Eighty-six studies with 3593 athletes were included in the null model. The forest plot can be found in Appendix A. The Bayesian meta-analysis showed a statistically significant effect [μ(SMD): 0.5, 95% CI: 0.34–0.67; HDI: 0.34–0.67; BF: 12.41], with moderate between-study heterogeneity and low within-study heterogeneity [within I2: 34.06%, between I2: 65.94%]. Imagery practice can effectively improve athletes’ performance. Figure 3 shows the cumulative probability and posterior density distribution of SMD, τ_within_ and τ_between_ in athletic performance.

### 3.6. Performance Model

In the performance model, 28 results on different sports performance indicators were presented, as shown in Figure 4 of the forest plot. However, only four indicators of athletic performance showed statistical significance, including agility [μ(SMD): 0.86, 95% CI: 0.18–1.55; HDI: 0.17–1.53; BF: 1.41], muscle strength [μ(SMD): 0.66, 95% CI: 0.13–1.18; HDI: 0.12–1.17; BF: 1.01], tennis performance [μ(SMD): 0.9, 95% CI: 0.41–1.4; HDI: 0.39–1.38; BF: 18.94] and soccer performance [μ(SMD): 0.63, 95% CI: 0.15–1.12; HDI: 0.16–1.12; BF: 1.52], with low within-study heterogeneity and moderate between-study heterogeneity [within I2: 9.5%, between I2: 57.79%].

### 3.7. Imagery Model

In the imagery model, 13 pairs of comparisons were included. The forest plot can be found in Figure 5. Only four pairs of comparisons showed statistical significance, including imagery practice vs. no practice [μ(SMD): 0.5, 95% CI: 0.31–0.68; HDI: 0.31–0.69; BF: 4480], imagery practice vs. PSTP [μ(SMD): −0.8, 95% CI: −1.12 to −0.48; HDI: −1.12 to −0.48; BF: 327.4], PSTP vs. feedback [μ(SMD): 1.53, 95% CI: 1.05–2.01; HDI: 1.04–2; BF: 28,300] and PSTP vs. no practice [μ(SMD): 0.95 95% CI: 0.66–1.23; HDI: 0.66–1.22; BF: 15,900], with low within-study heterogeneity and high between-study heterogeneity [within I2: 2.11%, between I2: 80.29%].

### 3.8. Athlete Model

In the athlete model, 27 types of athlete results were included. The forest plot can be found in Figure 6. Only two types of athletes showed statistical significance, including tennis players [μ(SMD): 1.12, 95% CI: 0.67–1.57; HDI: 0.68–1.57; BF: 663.9] and soccer players [μ(SMD): 0.58, 95% CI: 0.17–1.01; HDI: 0.17–1.01; BF: 1.96], with low within-study heterogeneity and moderate between-study heterogeneity [within I2: 10.1%, between I2: 53.89%].

### 3.9. Moderation Analysis

Seven moderators regarding athletes’ characteristics, including age, gender, training experience (in years), disease status, competitive level, sport type (team or individual), and PST experience, and four moderators concerning imagery dose and PSTP dose encompassing imagery practice duration (in days), frequency (sessions per week), intensity (minutes per session) and the number of PST in PSTPs were included in the moderation analysis. Regarding the moderation analysis of population characteristics, the charts can be found in Appendix A. The detailed results of the moderation analysis by population characteristics can be found in Table 1. The results in the gender model show that imagery practice had a worse effect on women than men (estimates: −0.01, −0.01 to −0.001). In the disease model, imagery practice had a better effect on intervening in healthy athletes (estimates: 0.42, 0.33–0.52). In the competitive level model, imagery practice is more effective for amateur athletes (estimates: 0.54, 0.35–0.74). In the sport type model (team or individual), imagery practice had a stronger effect on individual sports (estimates: 0.51, 0.38–0.65). In the PST experience model, imagery practice was more effective for athletes without psychological skills training experience (estimates: 0.45, 0.22–0.69).

The optimal modality appeared to be moderated by PST experience. Compared with the base model (R^2^ = 0.66), R^2^ was higher for the PST experience model (R^2^ = 0.76), training years model (R^2^ = 0.69), and the competitive level model (R^2^ = 0.67). R^2^ in the disease model (R^2^ = 0.66) was the same as that of the base model. R^2^ in the gender model (R^2^ = 0.59) was lower than in the base model. The R^2^ density plot can be found in Appendix A.

Four moderators regarding the imagery dose were added to the moderation analysis.

Regarding the duration of imagery practice, 100 days of imagery practice [coefficient estimates: 0.62, 95% CI: 0.28–0.93] were more effective than 50 days [coefficient estimates: 0.47, 95% CI: 0.32–0.61] and 20 days of practice [coefficient estimates: 0.38, 95% CI: 0.23–0.54]. The regression plot can be found in Figure 7.

Regarding the weekly frequency of imagery practice, the effect of practicing three times a week [coefficient estimates: 0.59, 95% CI: 0.42–0.75] was better than practicing once a week [coefficient estimates: 0.09, 95% CI: −0.24 to 0.42] or seven times a week [coefficient estimates: −0.13, 95% CI: −0.85 to 0.55]. The regression plot can be found in Figure 8.

Regarding the intensity of imagery practice, practicing for ten minutes at a time [coefficient estimates: 0.51, 95% CI: 0.34–0.7] was more effective than practicing for twenty [coefficient estimates: 0.22, 95% CI: −0.02 to 0.45], thirty [coefficient estimates: 0.26, 95% CI: −0.02 to 0.53], or forty-five minutes [coefficient estimates: 0.93, 95% CI: −0.1 to 2.11]. The regression plot can be found in Figure 9.

Regarding the number of PSTs included in PSTPs, the effect of combining three psychological skills trainings [coefficient estimates: 0.82, 95% CI: 0.37–1.37] was better than that of two [coefficient estimates: 0.55, 95% CI: 0.31–0.79] or four [coefficient estimates: 0.73, 95% CI: 0.11–1.37]. The regression plot can be found in Figure 10.

Because the model of the number of PSTs included in PSTPs only included data from PSTPs, R^2^ of this model was excluded from comparison with other models. Thus, the optimal modality appeared to be moderated by the weekly frequency. Compared with the base model (R^2^ = 0.55), R^2^ was higher for the duration model (R^2^ = 0.61), the weekly frequency model (R^2^ = 0.62), and the intensity model (R^2^ = 0.59). The R^2^ density plot can be found in Appendix A. The detailed results of the moderation analysis by imagery dose can be found in Table 2.

### 3.10. Quality Grade in Each Outcome

The quality grade in athletic performance was based on the sample size, results of the meta-analysis, and quality assessment to assess the risk of bias, inconsistency of results, indirectness, and imprecision of effect estimates. The results showed that the quality grade in the results of athletic performance was low because of the potential risk of bias and moderate heterogeneity. The plot can be found in Figure 11.

### 3.11. Publication Bias

The funnel plots exhibited asymmetry, suggesting a potential risk of bias. The majority of studies were non-affirmative and predominantly clustered on the left side of the graph. Notably, when only nonsignificant studies were included, the pooled effect size became statistically insignificant. The multilevel Egger’s test indicated significance (F_1,282_ = 10.36, *p* = 0.0014). However, the S-value results indicated that no plausible level of publication bias would nullify the observed effect (S-value = not possible). The funnel plot is presented in Figure 12.

## 4. Discussion

This is the first Bayesian multilevel meta-analysis identifying the optimal dosage and moderated effects of imagery practice on athletic performance. This systematic review and meta-analysis consolidates information about the impact of (1) imagery practice alone or in conjunction with other psychological skills training (PST) on athletic performance, and (2) the dosage of imagery on athletes’ performance.

### 4.1. Summary of the Findings

The results suggest that both imagery practice alone and its combination with other psychological training can enhance athletes’ performance. Moreover, integrating imagery practice with additional psychological techniques yields greater benefits compared to using imagery practice or other single psychological training methods, such as video observation and feedback, in isolation. In the performance model, imagery practice could effectively improve athletes’ agility, muscle strength, tennis performance, and soccer performance. In the athlete model, the results showed that imagery practice was only effective for tennis players and soccer players. In the moderation analysis, 100 days, three times a week, with approximately ten minutes of imagery practice each time, had a better effect on improving athletes’ performance. Additionally, imagery practice was more effective for male amateur athletes who were healthy and had no prior experience with psychological training.

### 4.2. The Effect of Imagery Practice on Athletic Performance

Data analysis demonstrated that imagery practice positively influences sports performance, particularly in agility, muscle strength, tennis, and soccer performance. Furthermore, imagery practice appears to positively impact tennis and soccer athletes. The enhancement of agility is crucial for athletes, particularly in sports such as basketball and football, which necessitate swift behavioral adaptations to environmental fluctuations ([5]). During imagery practice, optimal synaptic connections are activated, playing a crucial role in enhancing agility during movement ([98]). Several studies gave supportive evidence ([29]; [62]; [72]; [86]). [62] ([62]) concluded that, in comparison to no practice, imagery practice can significantly enhance athletes’ reaction speed; nonetheless, it appears to be less beneficial than physical training. Moreover, evidence indicates that the combination of imagery practice and physical exercise yields greater enhancements in athletes’ agility compared to imagery practice in isolation ([55]; [85]). This suggests that integrating imagery practice with physical exercise may yield greater positive effects. However, this meta-analysis did not specifically investigate this aspect, as the primary focus of the study was to determine the effectiveness of imagery practice and identify its optimal combination and dosage.

Some research offers evidence that imagery practice can enhance athletes’ muscle strength ([2]; [21]; [26]; [35]; [51]). The effectiveness of imagery practice is seemingly contingent upon the surface size of the relevant cortical area associated with the muscle in the primary motor cortex ([51]). Paravlic et al. concluded that imagery practice is an effective approach for enhancing muscle strength in healthy adults. However, whether combining imagery practice with physical exercise is more beneficial than imagery practice alone remains unclear ([70]). This meta-analysis did not find significant improvements in lower limb strength through imagery practice. Some studies suggest that while imagery practice may not significantly enhance jump performance in athletes, there is a noticeable trend toward improvement ([23]; [69]). Therefore, even in the absence of statistically significant differences, subtle enhancements in technique remain valuable.

The results in the athlete model and the performance model demonstrated that imagery practice significantly enhances performance in tennis and soccer. A multitude of studies arrived at a comparable affirmative conclusion ([3]; [12]; [30]; [37]; [68]; [79]). Some evidence gives the explanation that interior images may yield valuable insights for angle estimation, which is essential for tennis serve performance ([32]). Robin and Dominique determined that imagery practice can substantially enhance tennis performance, encompassing forehand, backhand, volley quality, and service efficacy; however, there is a deficiency of further research on imagery practice specifically for female tennis players ([80]). Furthermore, internal imagery appears to be particularly beneficial for tennis players, with several scholars supporting its effectiveness in enhancing performance in closed, goal-directed tasks ([24]). This provides a robust explanation for the positive effectiveness of imagery practice in tennis performance.

On the other hand, imagery practice has also been shown to enhance soccer performance. Rhodes et al. found that imagery is effective in improving penalty kick accuracy. However, while the PETTLEP model does not support long-term improvements, the motivational approach of functional imagery training (FIT) has been found to facilitate sustained performance enhancements ([77]), and FIT can also enhance soccer athletes’ grit ([76]). FIT appears to be a suitable imagery intervention for soccer players. However, some evidence suggests that PETTLEP imagery practice can enhance goalkeepers’ spatial anticipation during penalty kicks ([3]). Moreover, studies have shown that cognitive-specific imagery interventions can enhance soccer skill performance in youth athletes, with younger athletes experiencing greater benefits from imagery training ([65]). Therefore, while we can currently confirm that imagery practice can improve soccer players’ performance, further research is needed to identify the optimal type of imagery practice. Additionally, the lack of statistical significance in other athletic performance results does not imply that imagery practice is entirely ineffective—such as in sprinting. Even a one-second difference in sprint times can be significant, so the small improvements that imagery practice may offer in other areas should not be overlooked. Some evidence suggests that imagery practice can enhance repetitive sprinting ability ([86]).

Combining imagery practice with other psychological training has been shown to be significantly more effective than imagery practice alone. Several studies provide supporting evidence for this approach ([9]; [41]; [78], [81]). Research has shown that the combined use of self-talk and imagery practice can enhance athletes’ motor skills, improve athletic performance ([41]), and significantly increase muscle strength, particularly in kickboxers ([91]). Gmamdya et al. concluded that the integration of imagery practice with feedback, video observation, and regular physical exercise significantly enhances athletic performance ([33]). However, recent studies have suggested that combining virtual reality with imagery practice has a more significant impact on shooting performance and muscle activation than the combination of imagery practice and video observation ([9]; [10]). Therefore, combining imagery practice with various psychological skills training appears to produce inconsistent beneficial effects. Virtual reality (VR), as an emerging and promising technology, has garnered significant attention in the field of sports. More research is needed in the future to explore the most effective imagery practice methods tailored to athletes.

### 4.3. The Dose–Response Relationship of Imagery Practice and Athletic Performance

The results indicated that imagery practice lasting around ten minutes, three times a week, over the course of one hundred days, had the most significant positive impact on athletes’ performance. Several studies provide supporting evidence for this. From a broader perspective, Driskell et al. suggested that the benefits of mental practice diminish with longer durations, and that an optimal practice length of approximately 20.8 min is ideal for PST ([27]). Secondly, regarding imagery practice, a meta-analysis suggested that increasing the number of sessions enhances the effectiveness of imagery practice for athletes ([89]), and four weeks, three times a week, fifteen minutes of imagery practice each time are more beneficial for improving muscle strength in healthy adults ([70]). This aligns with our results; however, it appears that the optimal dosage of imagery practice may depend on the type of motor performance. For instance, the ideal imagery intervention duration for muscle development in healthy adults seems to be around four weeks, which differs from the overall dosage recommended in our study.

Moreover, our findings indicated that the integration of imagery practice with additional psychological training yielded superior outcomes, and it was optimal to combine one or two types of psychological training to achieve the most effective results. The most popular combinations of imagery practice include imagery practice with self-talk ([22]; [78], [81]), video observation ([23]; [21]; [52]; [63]; [88]), video observation and feedback ([33]), and VR ([9]). However, the optimal combination of imagery practice remains unclear, and further research is needed to provide solid and robust evidence in the future.

Finally, through moderation analysis, we found that imagery practice seems to have a poor effect on female athletes. Some studies provide supporting evidence ([13]; [36]; [90]). Robin et al. concluded that in the current research on imagery practice, the representation of female athletes is relatively limited ([80]). Consequently, subsequent research should emphasize gender differences in imagery practice to yield more substantial and comprehensive conclusions.

### 4.4. Strengths, Limitations, and Future Directions

This was the inaugural Bayesian multilevel meta-analysis that comprehensively examined the ideal athlete groups suitable for imagery practice, identified the specific performance metrics that can be enhanced through imagery practice, and delineated the optimal dosage of imagery practice appropriate for this athlete population. It provided more substantial and thorough evidence for the prospective implementation of psychological skills training in sports.

However, several limitations must be addressed. First, there is moderate between-study heterogeneity in the null model, with an overall between-study heterogeneity of around 60%. This heterogeneity decreased somewhat in the performance and imagery models, suggesting that it was partly due to differences in performance indicators and partly due to variations in imagery intervention types. The failure to classify the types of imagery practice in this study contributed to some of the between-study heterogeneity, which in turn reduced the reliability of the results. Second, there was some degree of publication bias, which diminished the quality and reliability of the findings. Third, the inclusion of numerous small-sample studies (*n* < 20) may have contributed to both heterogeneity and publication bias. Additionally, certain athlete groups, such as CrossFit athletes and equestrian athletes, were underrepresented. Fourth, we focused solely on the dosage of imagery practice and did not explore the optimal dosage when combining multiple psychological training methods. Fifth, we did not differentiate between the types of imagery interventions, and the optimal type of imagery practice remains unknown. Sixth, in the dose analysis, a trend toward positive effects was observed with imagery practice lasting over forty minutes. However, due to a lack of further research, the effectiveness of imagery practice lasting more than forty minutes remains uncertain.

Finally, although weekly informative priors were used, effect estimates might still have been influenced by prior specifications. Future work may consider sensitivity analyses with varying priors. High-quality randomized controlled trials with large sample sizes should be conducted in the future to further establish robust evidence on the effects of imagery practice on athletic performance and its optimal dosage. Moreover, exploring the integration of VR-based imagery or AI-guided PST customization may further enhance training personalization and transferability.

## 5. Conclusions

This study provides compelling evidence that imagery practice significantly enhances athletic performance, particularly in agility, muscular strength, and in tennis and soccer domains. The integration of imagery with one or two psychological skills trainings outperforms standalone imagery. Notably, the optimal dosage—ten minutes per session, three times per week, over 100 days—demonstrated the most robust effect. Through a Bayesian multilevel meta-analysis, this research contributes methodologically rigorous insights into the effect of heterogeneity and moderator structures of psychological skills training. The findings of this study support our hypotheses, demonstrating that imagery practice enhances athletic performance and that a longer duration of practice yields more pronounced effects.

These findings hold direct implications for designing personalized mental training programs, particularly for healthy, amateur athletes without prior PST experience. Coaches and sport psychologists are encouraged to adopt structured, time-efficient imagery protocols informed by this dosage evidence.

Future research should explore (a) the interaction between imagery modalities and motor task types, (b) comparative effectiveness of various PST combinations, and (c) gender-based differences in psychological responsiveness.

## Figures and Tables

**Figure 1 behavsci-15-00685-f001:**
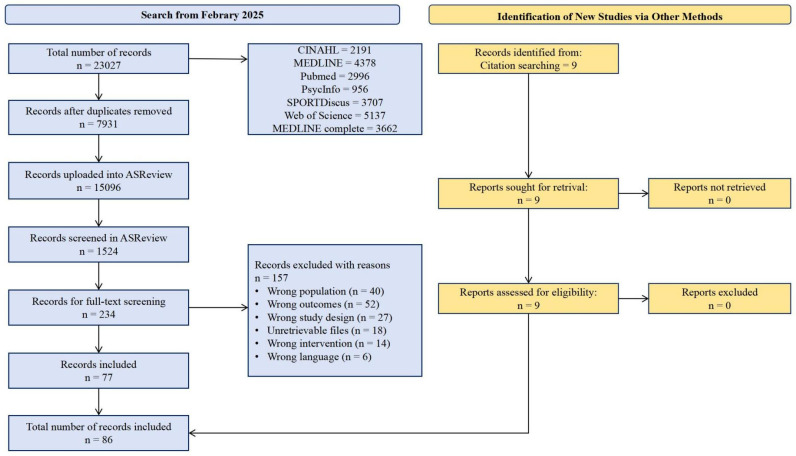
PRISMA flow chart for the identification of the included studies.

**Figure 2 behavsci-15-00685-f002:**
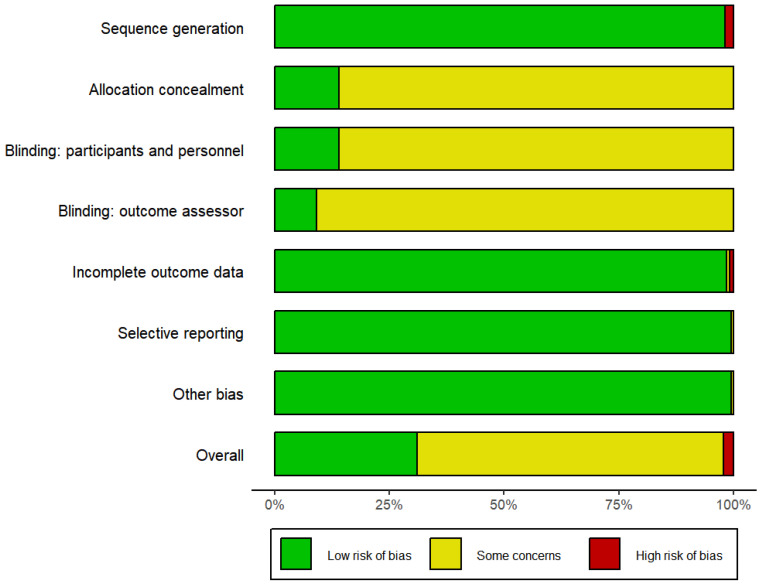
Risk-of-bias summary.

**Figure 3 behavsci-15-00685-f003:**
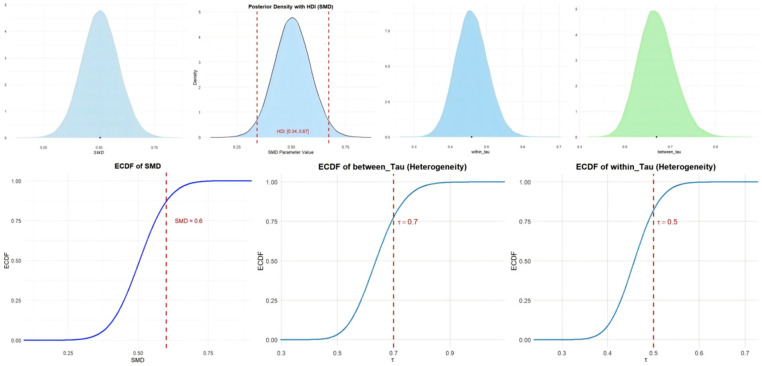
The cumulative probability, HDI distribution, and posterior density distribution (SMD, τ_within_ and τ_between_) in athletic performance.

**Figure 4 behavsci-15-00685-f004:**
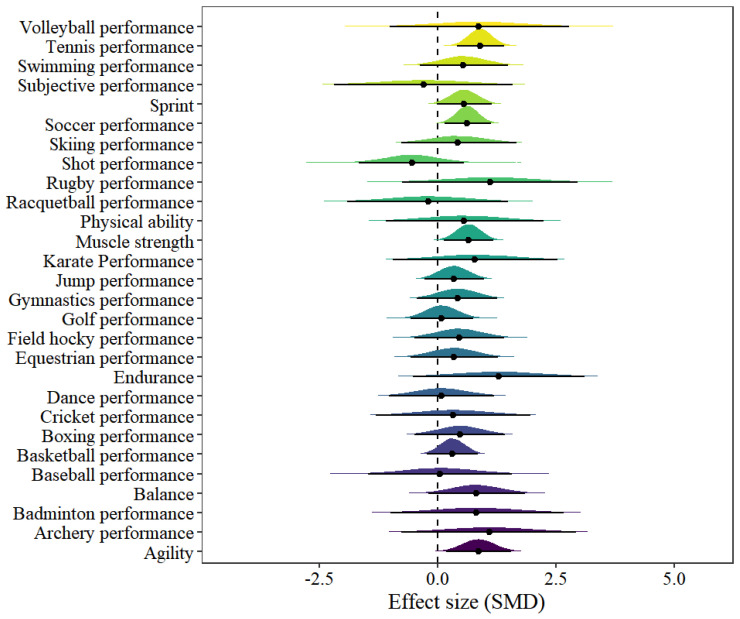
The forest plot in the performance model.

**Figure 5 behavsci-15-00685-f005:**
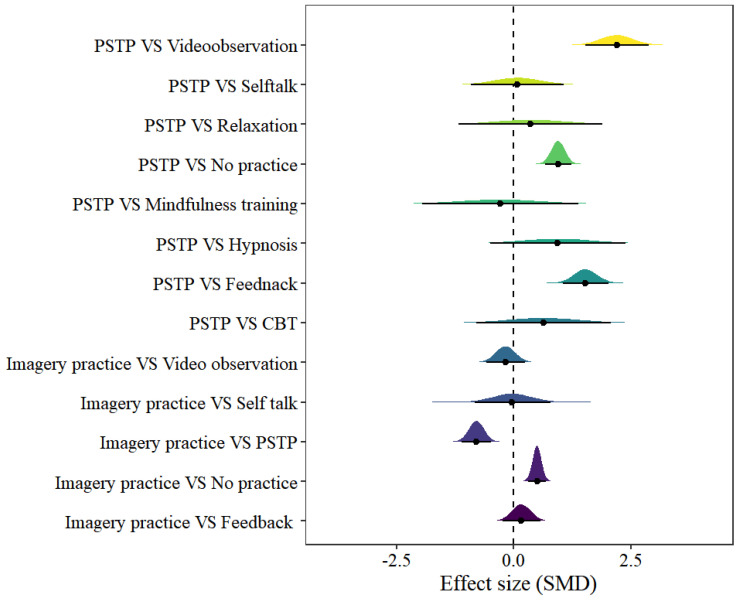
The forest plot in the imagery model.

**Figure 6 behavsci-15-00685-f006:**
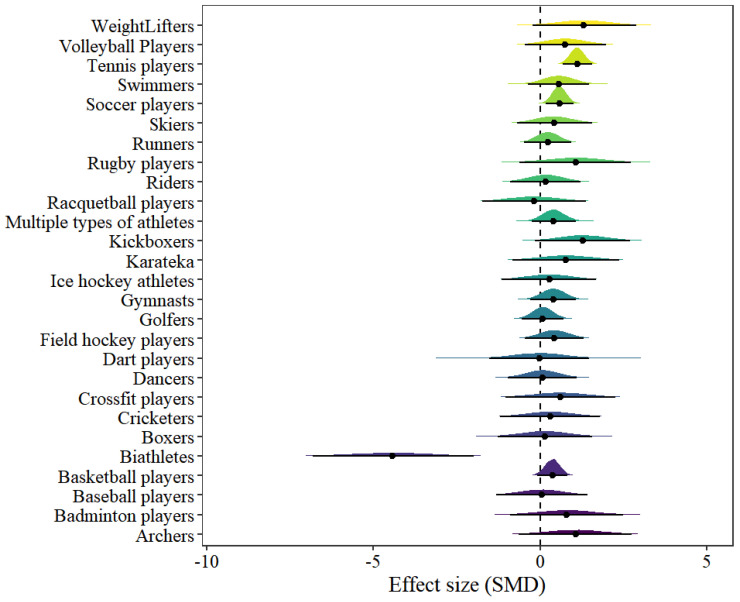
The forest plot in the athlete model.

**Figure 7 behavsci-15-00685-f007:**
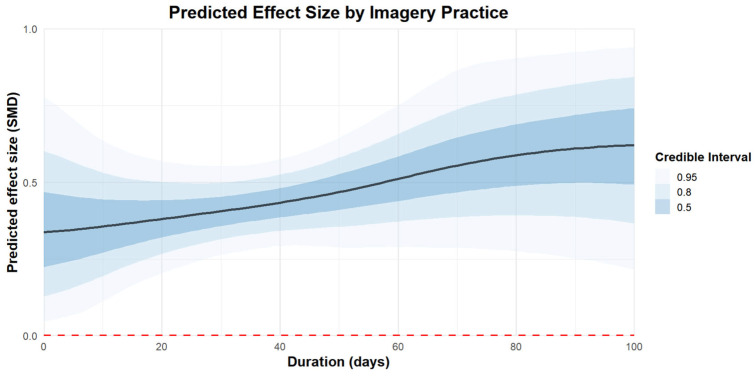
The moderation analysis by duration (days).

**Figure 8 behavsci-15-00685-f008:**
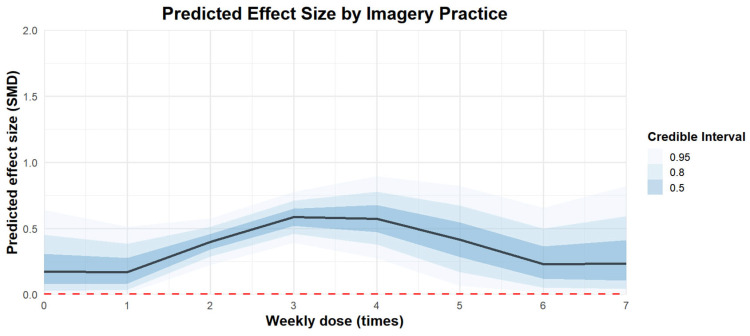
The moderation analysis by weekly frequency (times).

**Figure 9 behavsci-15-00685-f009:**
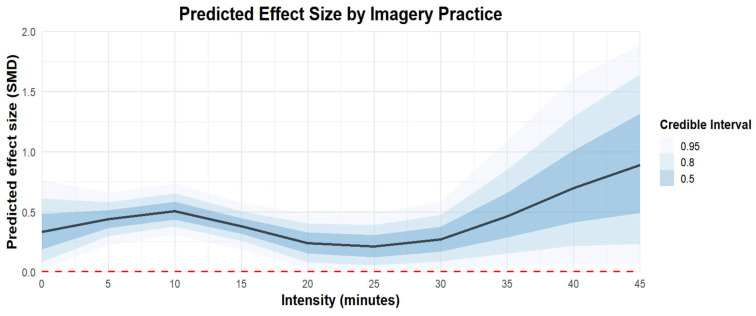
The moderation analysis by intensity (minutes).

**Figure 10 behavsci-15-00685-f010:**
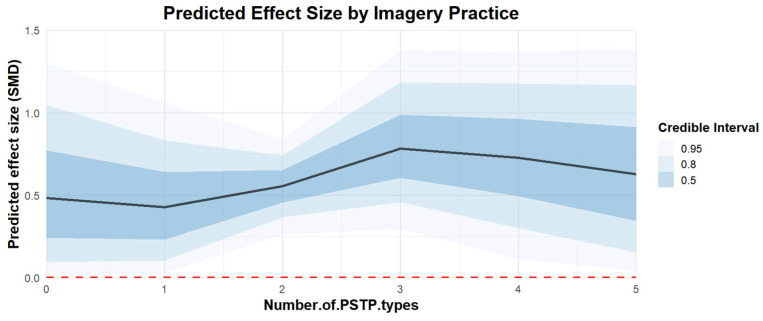
The moderation analysis by the number of PSTs in PSTP.

**Figure 11 behavsci-15-00685-f011:**
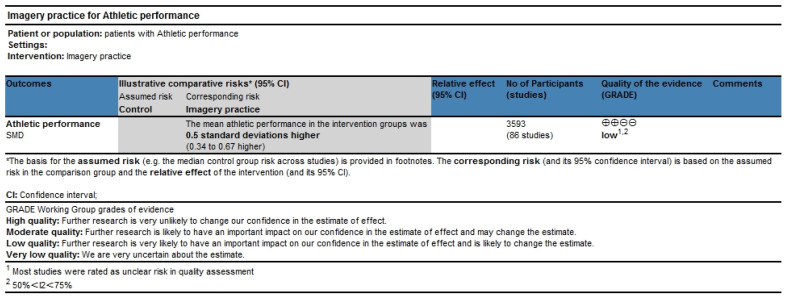
Quality grade of athletic performance. ⊕ indicates a positive rating or higher quality of evidence; ⊖ indicates a negative rating or lower quality of evidence.

**Figure 12 behavsci-15-00685-f012:**
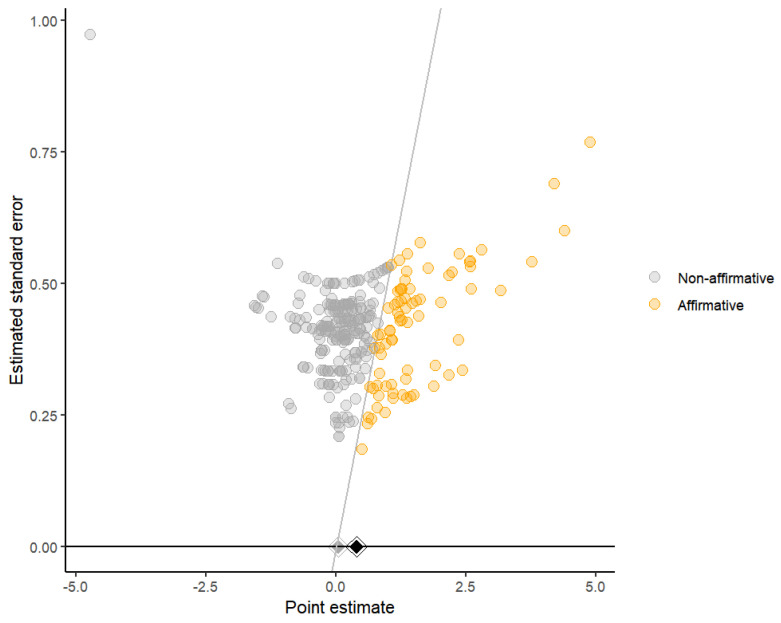
Funnel plot of athletic performance.

**Table 1 behavsci-15-00685-t001:** Detailed results of the moderation analysis by population characteristics.

	Moderator	Estimate	l-95% CI	u-95% CI	R^2^	l-95% CI	u-95% CI
1	Gender	−0.01	−0.01	−0.001	0.59	0.5	0.69
2	Age	0.01	−0.02	0.02	0.65	0.56	0.73
3	Training years	−0.02	−0.08	0.03	0.69	0.52	0.82
4	Disease (TRUE)	0.86	−0.62	2.37	0.66	0.58	0.73
5	Disease (FALSE)	0.42	0.33	0.52	0.66	0.58	0.73
6	Competitive level (amateur)	0.54	0.35	0.74	0.67	0.59	0.74
7	Competitive level (elite)	−0.14	−0.37	0.09	0.67	0.59	0.74
8	Competitive level (mix)	−0.49	−1.03	0.05	0.67	0.59	0.74
9	TI (individual sports)	0.51	0.38	0.65	0.66	0.58	0.73
10	TI (team sports)	−0.15	−0.36	0.04	0.66	0.58	0.73
11	TI (mix)	−0.32	0.7	0.05	0.66	0.58	0.73
12	PST experience (TRUE)	−0.12	−1.09	0.86	0.76	0.62	0.87
13	PST experience (FALSE)	0.45	0.22	0.69	0.76	0.62	0.87
14	Base	0.43	0.33	0.52	0.66	0.58	0.73

PST: psychological skills training; TI: team or individual sports.

**Table 2 behavsci-15-00685-t002:** Detailed results of the moderation analysis by imagery dose.

	Moderator	Type	Estimate	l-95% CI	u-95% CI	R^2^	l-95% CI	u-95% CI
1	Duration	20 days	0.38	0.23	0.54	0.61	0.48	0.72
2	Duration	50 days	0.47	0.32	0.61	0.61	0.48	0.72
3	Duration	100 days	0.62	0.28	0.93	0.61	0.48	0.72
4	Weekly frequency	Once a week	0.09	−0.24	0.42	0.62	0.47	0.75
5	Weekly frequency	Three times a week	0.59	0.42	0.75	0.62	0.47	0.75
6	Weekly frequency	Seven times a week	−0.13	−0.85	0.55	0.62	0.47	0.75
7	Intensity	10 min	0.51	0.34	0.7	0.59	0.45	0.72
8	Intensity	20 min	0.22	−0.02	0.45	0.59	0.45	0.72
9	Intensity	30 min	0.26	−0.02	0.53	0.59	0.45	0.72
10	Intensity	45 min	0.93	−0.1	2.11	0.59	0.45	0.72
11	PSTP types	2 PSTs	0.55	0.31	0.79	0.79	0.67	0.88
12	PSTP types	3 PSTs	0.82	0.37	1.37	0.79	0.67	0.88
13	PSTP types	4 PSTs	0.73	0.11	1.37	0.79	0.67	0.88
14	Base	NA	0.36	0.27	0.46	0.55	0.43	0.66

PSTP: psychological skills training package.

## Data Availability

No new data were created or analyzed in this study.

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
