# Peer review of "The Effects of Imagery Practice on Athletes’ Performance: A Multilevel Meta-Analysis with Systematic Review"

_behavsci, 2025, doi:10.3390/bs15050685_

Round 1
Reviewer 1 Report
Comments and Suggestions for Authors
- The manuscript, 'The Effects of Imagery Practice on Athletes’ Performance: A 2 Multilevel Meta-analysis with Systematic Review' reflects sound conception of the research, execution and reporting, with great recommendations for further study.
- Nevertheless, Page 2, second paragraph lines 58 to 61 can be corrected to remove repetitions as below:
"motor skill performances, including basketball (Blumenstein et al., 2018; Fazel et al., 2018), gymnastics (Battaglia et al., 2014; Khalaf et al., 2024), tennis (Laurent et al., 2024; Nicolas et al., 2025), soccer (Amini Farsani et al., 2023; Robin et al., 2020), among others."
-
The ending sentence on page 2, paragraph two, lines 65 and 66 gives the impression that prior research findings are overwhelmingly supportive of the effectiveness of imagery practice, "Therefore, numerous studies have provided substantial evidence supporting the effective- ness of imagery practice in enhancing athletic performance." it would be best to frame it to reflect that previous studies had flaws hence the need for more robust investigation as laid out in the subsequent paragraph.
- Line 75 on page 2, PETTLEP should follow use of full name for the reader to follow. Put full name and then abbreviation in brackets.
- On page 3, under eligibility, the tense should be past and not future.....instead of 'will be', it should be 'were'..this is in two places.
- Page 3 last paragraph and page 4 first paragraph have "performance" monotonously repeated. Once performance is in the stem, the motor areas can just stand alone without adding performance.
- Page 5, under "2 Imagery model", the tense again is future when it should be past...
- Page 5, line 230 has a 'Furthermore' as a sentence. Could be deleted or written in full to make sense.
- Page 17. line 485, "Moreover, studies (and not study)....
- The study is robust and deserving of publication after the corrections.
Author Response
- The manuscript, 'The Effects of Imagery Practice on Athletes’ Performance: A 2 Multilevel Meta-analysis with Systematic Review' reflects sound conception of the research, execution and reporting, with great recommendations for further study.
Reply: I sincerely appreciate your valuable comments and suggestions. I will go through each of them carefully and respond to all the points with detailed revisions.
- Nevertheless, Page 2, second paragraph lines 58 to 61 can be corrected to remove repetitions as below:
"motor skill performances, including basketball (Blumenstein et al., 2018; Fazel et al., 2018), gymnastics (Battaglia et al., 2014; Khalaf et al., 2024), tennis (Laurent et al., 2024; Nicolas et al., 2025), soccer (Amini Farsani et al., 2023; Robin et al., 2020), among others."
Reply: Thank you for your insightful comment. The revision has been made accordingly in the relevant section.
- The ending sentence on page 2, paragraph two, lines 65 and 66 gives the impression that prior research findings are overwhelmingly supportive of the effectiveness of imagery practice, "Therefore, numerous studies have provided substantial evidence supporting the effectiveness of imagery practice in enhancing athletic performance." it would be best to frame it to reflect that previous studies had flaws hence the need for more robust investigation as laid out in the subsequent paragraph.
Reply: Thank you for your insightful comment. In response, we have revised the concluding sentence of the second paragraph on page 2 (lines 65–66) to adopt a more balanced and critical tone. The new sentence now reads: “Thus, while the existing evidence suggests that imagery practice holds promise, the methodological constraints and limited scope of past studies highlight the need for more rigorous and comprehensive investigations.”
Additionally, to improve the logical flow, we removed the word “However” at the beginning of the subsequent paragraph. No other modifications were made to that paragraph.
- Line 75 on page 2, PETTLEP should follow use of full name for the reader to follow. Put full name and then abbreviation in brackets.
Reply: Thank you for pointing this out. We have revised line 75 on page 2 to provide the full name of the PETTLEP model upon its first mention. The updated sentence now reads: “Imagery based on the Physical, Environment, Task, Timing, Learning, Emotion, and Perspective (PETTLEP) model is commonly used in sport settings...”
- On page 3, under eligibility, the tense should be past and not future.....instead of 'will be', it should be 'were'..this is in two places.
Reply: Thank you for your feedback. We have revised the hypotheses section to reflect the past tense as suggested. Specifically, we changed the tense from future to past in all hypotheses to accurately reflect the completed nature of the study. The updated sentences now read:
“Imagery practice had a positive effect on athletic performance…”
“It was most effective for tennis and soccer players…”
“Engaging in imagery practice three times weekly for approximately 10 minutes over 100 days improved performance…”
“The efficacy of imagery practice was greater among male athletes…”
Importantly, following this initial revision, we further refined the structure of the research hypotheses based on feedback from other reviewers. Specifically, the hypotheses were consolidated into two main statements:
(1) Imagery practice had a positive effect on the motor areas; and
(2) The effectiveness of imagery practice increased with longer intervention duration.
- Page 3 last paragraph and page 4 first paragraph have "performance" monotonously repeated. Once performance is in the stem, the motor areas can just stand alone without adding performance.
Reply: Thank you for your comment. In response, we have revised the last paragraph on page 3 and the first paragraph on page 4 to avoid repetitive use of the term “performance.” We have substituted "performance" with related terms, such as "motor areas," to maintain clarity and variation.
- Page 5, under "2 Imagery model", the tense again is future when it should be past...
Reply: Thank you for your comment. We have corrected the tense in the section under "2 Imagery model" on page 5. Specifically, we replaced the future tense "will be made" with the past tense "was made" to accurately reflect the study's completed nature.
- Page 5, line 230 has a 'Furthermore' as a sentence. Could be deleted or written in full to make sense.
Reply: Thank you for your suggestion. We have deleted the word "Furthermore" on page 5, line 230 to ensure the sentence flows more naturally and maintains coherence.
- Page 17. line 485, "Moreover, studies (and not study)....
Reply: Thank you for your feedback. We have corrected the text on page 17, line 485, by changing "study" to "studies" to reflect the plural form.
The study is robust and deserving of publication after the corrections.
General Response:
Thank you for all your valuable suggestions and comments. I have addressed all the issues raised and made the necessary revisions. If you have any further questions or suggestions, please feel free to let me know, and I will make additional modifications as needed.
Reviewer 2 Report
Comments and Suggestions for Authors
The article is great opportunity and high quality assessment for reviewing the effects of different psychological practices on athlethes' performance.
Generally, the article follows a well-built methodology with application of many current (also AI based) technologies, which can provide a scientific guideline for how to build up a review.
However, I have some major concerns about some points, which definitely need to be addressed before publication:
- Hypotheses do not seem to come from the introduction, rather it seems, that the hypothesis is based on what is found as a result from the results section. Hypotheses must be built before seeing the results, and here there is no explanation why
- why engaging in imagery practice seem to be more effective than other types of practices
- why three times weekly for approximately 10 minutes over 100 days will be the optimal (it contradicts the part of introduction: Recent studies indicate that imagery practice for four weeks, lasting fifteen minutes per session, three times weekly, appears to yield the most beneficial results for healthy individuals... (Itoh et al., 2023; Paravlic et al., 2018))
- why the highest effect is expected for tennis and soccer players ( in introduction it is written: imagery practice appears to positively influence the enhancement of specialized motor skills, including basketball performance (Blumenstein et al., 2018; Fazel et al., 2018), gymnastics performance (Battaglia et al., 2014; Khalaf et al., 2024), tennis performance (Laurent et al., 2024; Nicolas et al., 2025), soccer performance (Amini Farsani et al., 2023; Robin et al., 2020), among others.))
- It is not clear what PSTP types were examined and what PSTP types mean (e.g. see Table 2).
- It is not clear what 2,3,4 PSTP types mean (I assume it means more types of PTSP - it should be clearly stated what types of exercises are meant here - as they seem to be more effective, than imaginary exercise)
- In comparing effect imaginary practice vs PSTP, PSTP seems to have higher effect. Mindfulness, relaxation and self-talk seem to have similar effect as PSTP. According to this, logically it comes that imaginary practice is less effective than PSTP, mindfulness, relaxation... How can we conclude that imaginary practice is suggested for athletes? It should be concluded that other types of psychological practices are more effective ( IT SHOULD BE MORE EMPHASIZED!) and if more types of PTSP are applied than it is more effective (IT SHOULD BE EXAMINED MORE DEEPLY WHICH PTSP ARE EFFECTIVE).
- 45 minutes seem to be more effective than 10 minutes - so we cannot conclude that 10 min is optimal.
In sum, it is a very important and big work, but consequences drawn from the results seem to be biased and it needs to be objectively and more carefully explained what kind of results can be drawn. Also, the hypotheses must be based on a priori explanation (theoreticel deduction) and not forming the results into a hypothesis.
Author Response
The article is great opportunity and high quality assessment for reviewing the effects of different psychological practices on athlethes' performance.
Generally, the article follows a well-built methodology with application of many current (also AI based) technologies, which can provide a scientific guideline for how to build up a review.
Reply: Thank you very much for your detailed comments and suggestions on my manuscript. I will respond to each of your points one by one below.
However, I have some major concerns about some points, which definitely need to be addressed before publication:
- Hypotheses do not seem to come from the introduction, rather it seems, that the hypothesis is based on what is found as a result from the results section. Hypotheses must be built before seeing the results, and here there is no explanation why
Reply: Thank you for pointing this out. Initially, before conducting the meta-analysis, our hypothesis was that imagery practice can improve athletic performance. However, due to the broad and multifaceted nature of athletic performance, we were unsure which specific components might benefit the most, and previous studies typically focused on one or two aspects (e.g., agility or tennis performance), without providing a comprehensive summary. During manuscript preparation, we articulated our hypotheses more precisely based on theoretical expectations and the structure of the existing literature, which may have inadvertently aligned closely with the final results and appeared to deviate from the Introduction. In response to your comment, we have revised the hypothesis section to better reflect the initial research questions and the rationale grounded in prior studies, rather than the observed outcomes.
- why engaging in imagery practice seem to be more effective than other types of practices
Reply: Thank you for your insightful question. The primary aim of our study was to investigate the effectiveness of imagery practice on athletic performance. To expand on this, we also examined studies that implemented combined psychological skills training programs (PSTP), under the condition that imagery practice was included as a core component. This allowed us to explore whether imagery practice, when integrated with other psychological strategies such as self-talk or goal setting, might yield greater benefits than imagery alone.
Our results indicated that PSTP incorporating imagery practice were indeed more effective than using imagery practice alone. While we compared PSTP with other forms of psychological training (e.g., feedback, video observation), our findings did not suggest that imagery practice alone was superior to these other techniques. Rather, it was the combination of imagery with other psychological skills that demonstrated enhanced effectiveness.
We emphasize that the focus of our analysis remains on imagery practice—either as a standalone intervention or as part of a broader psychological training program. This aligns with our study's title and objectives. We have clarified this rationale in the revised manuscript to ensure the purpose is more explicitly communicated (2.8 moderation analysis: PSTP types refer to the number of psychological skills training integrated alongside imagery practice within a training package, categorized as combinations of two, three, or four techniques in total (i.e., imagery plus one, two, or three additional psychological strategies). )
- why three times weekly for approximately 10 minutes over 100 days will be the optimal (it contradicts the part of introduction: Recent studies indicate that imagery practice for four weeks, lasting fifteen minutes per session, three times weekly, appears to yield the most beneficial results for healthy individuals... (Itoh et al., 2023; Paravlic et al., 2018))
Reply: Thank you for your thoughtful comment. We acknowledge that our hypotheses were overly precise, which may have caused confusion. At the beginning of this meta-analysis, we speculated that a longer duration of imagery practice would generally lead to better performance, though we did not have a clear idea of the exact optimal dosage. After completing the meta-analysis, we identified that imagery practice three times per week, for approximately 10 minutes per session over a period of 100 days, was associated with the greatest performance gains across a broad range of sports.
This conclusion does not conflict with previous studies such as Itoh et al. (2023) or Paravlic et al. (2018). These prior works were focused on specific performance indicators or populations. For example, Itoh et al. examined basketball free throw accuracy and found 15-minute sessions more effective than shorter ones. Paravlic et al. studied muscular strength improvements in healthy adults. In contrast, our study synthesizes findings across various sports and performance domains, providing a more generalized and macro-level conclusion about the optimal dosage of imagery practice for athletes’ performance.
Moreover, our analysis focused specifically on athletic populations, whereas some earlier studies involved general adult populations. Given the differences in both outcome scope and participant characteristics, we believe these conclusions are not contradictory but rather complementary, reflecting different research perspectives.
Finally, the initial hypothesis—that longer durations of imagery practice would lead to better athletic performance—was based on prior meta-analytic findings, particularly Simonsmeier et al. (2018), rather than being arbitrarily proposed.
- why the highest effect is expected for tennis and soccer players ( in introduction it is written: imagery practice appears to positively influence the enhancement of specialized motor skills, including basketball performance (Blumenstein et al., 2018; Fazel et al., 2018), gymnastics performance (Battaglia et al., 2014; Khalaf et al., 2024), tennis performance (Laurent et al., 2024; Nicolas et al., 2025), soccer performance (Amini Farsani et al., 2023; Robin et al., 2020), among others.))
Reply: Thank you for your comment. The hypothesis that imagery practice would yield the highest effects for tennis and soccer players was not part of our original research hypotheses, but rather an observation that emerged from the meta-analysis results. We acknowledge that including this as a hypothesis in the introduction was misleading. Accordingly, we have revised the research hypotheses to reflect only those that were pre-specified prior to data analysis: (1) imagery practice enhances athletic performance; and (2) the longer the intervention duration, the greater the expected effect.
These two original hypotheses are indeed what I wanted to investigate at the beginning of conducting meta-analysis, and they will not conflict with the content of the Introduction
- It is not clear what PSTP types were examined and what PSTP types mean (e.g. see Table 2).
- It is not clear what 2,3,4 PSTP types mean (I assume it means more types of PTSP - it should be clearly stated what types of exercises are meant here - as they seem to be more effective, than imaginary exercise)
- In comparing effect imaginary practice vs PSTP, PSTP seems to have higher effect. Mindfulness, relaxation and self-talk seem to have similar effect as PSTP. According to this, logically it comes that imaginary practice is less effective than PSTP, mindfulness, relaxation... How can we conclude that imaginary practice is suggested for athletes? It should be concluded that other types of psychological practices are more effective ( IT SHOULD BE MORE EMPHASIZED!) and if more types of PTSP are applied than it is more effective (IT SHOULD BE EXAMINED MORE DEEPLY WHICH PTSP ARE EFFECTIVE).
Reply: Thank you for the comments. I will address all three questions collectively. First, in this context, "PSTP types" refers to the number of psychological skills training (PST) components included within a training package that incorporates imagery practice. During the literature screening process, we included all interventions that combined imagery practice with other PST elements, as our aim was to investigate whether integrating imagery with additional psychological strategies would yield greater benefits than employing imagery practice alone.
Our meta-analytic results demonstrated that combining imagery with other forms of psychological training produced superior effects compared to imagery alone. This raised a further question: among the studies that integrated imagery with other PSTs, some included one additional strategy, others included two, and some incorporated three or more. Thus, we conducted a moderation analysis to explore whether the number of combined PST components moderated the intervention effect.
The findings revealed that combining imagery practice with two additional psychological strategies (i.e., a total of three PST components) produced the most favorable outcomes. However, we did not further categorize or define specific PST types within these packages. This is because the current investigation is preliminary in nature, and the types of PST combined with imagery varied greatly across studies, with limited available data to support a more granular classification.
I mentioned the explanation of PSTP that includes imagery practice in the Method (1. The first type includes imagery practice that is not integrated with other psychological skills training, whereas the second type consists of psychological skills training packages (PSTP) that incorporate imagery practice. 2. incorporating eight moderator variables: age, gender, training experience (in years), disease status, competitive level, sport type (team or individual). PST experience, and the number of PST in PSTP (PSTP types).)
Furthermore, an additional explanation of PSTP types was included in Section 2.8 (moderation analysis) of the Methods to clarify their definition. (PSTP types refer to the number of psychological skills training integrated alongside imagery practice within a training package, categorized as combinations of two, three, or four techniques in total (i.e., imagery plus one, two, or three additional psychological strategies).
- 45 minutes seem to be more effective than 10 minutes - so we cannot conclude that 10 min is optimal.
Reply: Thank you for your valuable comment. Our findings indicate that a 10-minute imagery intervention demonstrates a more favorable effect compared to the 45-minute condition, as the moderation analysis for the 45-minute group yielded a non-significant result [coefficient estimates: 0.93, 95%CI: -0.1 to 2.11].
However, the regression plot revealed a positive trend, although the confidence interval crossed zero, rendering the result statistically non-significant. This non-significance may be attributable to the limited amount of data available for this condition. This limitation has been acknowledged in the discussion section (see 4.4.Strengths, Limitations, and Future Directions), where we also highlighted the need for future research to further investigate the effects of extended-duration imagery interventions.
- In sum, it is a very important and big work, but consequences drawn from the results seem to be biased and it needs to be objectively and more carefully explained what kind of results can be drawn. Also, the hypotheses must be based on a priori explanation (theoreticel deduction) and not forming the results into a hypothesis.
Reply: Thank you very much for your valuable comments. We have carefully revised and adjusted the manuscript accordingly. If there are any further suggestions, we would be glad to respond and make additional improvements.
Reviewer 3 Report
Comments and Suggestions for Authors
Dear researchers, I'm attaching a Word document with some comments.
It's an incredible piece of work, a very high-level meta-analysis.

Author Response
Dear researchers,
I'm sending you my comments,reflections,questions,and observations on the article:
First of all,I want to congratulate you on the work you've done.Your mastery of statistics is evident, and this is an incredible meta-analysis.To be honest,I was pleasantly surprised.The topic was also very interesting.
Now,I'd like to make some comments that I hope will help you improve your work.
Reply: Dear reviewer, I sincerely appreciate every comment and suggestion you have provided for my manuscript. I will carefully review and address each of them one by one, and provide a detailed response accordingly.
Introduction
- "However,notwithstanding the increasingdata endorsing imagery practice,considerable obstacles ersist in its execution”(line 67).For example?
Reply: Thank you very much for your comment. My intention was for the content following this sentence to support the central idea, which is why I did not include specific references in that particular sentence. However, each subsequent citation (e.g., Reinebo et al., 2024; Quinton et al., 2014; Avila et al., 2015; Paravlic et al., 2018; Simonsmeier et al., 2018) directly supports this point.
- Mustinclude what the acronym "PETTLEP"refers
Reply: Thank you for your comment. Another reviewer also raised this point, so I have included the full form of the PETTLEP acronym in the Introduction section (the Physical, Environment, Task, Timing, Learning, Emotion, and Perspective (PETTLEP) model).
- Simonsmeieret suggested that imagery practice appears to provide advantageous effects solely for the athletic performance of elite athletes(Simonsmeier et al.,2018).
I have a feeling the quote should read like this:Simonsmeier et al.(2020)suggested that imagery practice appears to provide advantageous effects solely for the athletic performance of elite athletes.
Reply: Thank you for the suggestion. I have already made the revision.
- "Prior meta-analysesoften overlook dosage parameters,rely on frequentist methods,and fail to account for nested data "Be careful with this type of statement;they don't cite any meta- analysis,and even if what they say is true,if no one declares it a limitation,it seems very subjective to me to declare it as such.
Reply: Thank you for this insightful comment. I have removed this sentence, as it was indeed a subjective statement.
- Ithink it's fantastic that you close the introduction with your hypotheses,but I think it's important to contextualize them and explain why you believe them,especially the one about soccer and tennis, and the one about the benefits only for You talked very little about these topics.
Reply: Thank you for your comment. Another reviewer also raised several concerns regarding the research hypotheses. Indeed, the original hypotheses I formulated prior to conducting this meta-analysis were: (1) imagery practice can enhance athletic performance, and (2) longer durations of imagery practice are associated with greater effects. However, as the results of the meta-analysis emerged, I attempted to refine the hypotheses to align more closely with the findings, which unintentionally caused confusion. Therefore, I have revised the research hypotheses to reflect the original assumptions made at the outset of the meta-analysis, prior to obtaining the results.
- Andas a suggestion,you could also explicitly add your main While it's clear from reading the final part of the introduction,it should be explicit,especially if you included the Prisma checklist and the element explicitly requests it in point 4.
Reply: Thank you for your valuable comment. We have revised the final paragraph of the Introduction to clearly articulate our research objectives.
Methods
Congratulations! A really good section of material and methods,very comprehensive,from the writing to the way you cite and justify everything.Very good work.
It's great that you've registered your work on "PROSPERO"
- Whydidn't they use Scopus?
Reply: Thank you for your question. We initially included Scopus in our search strategy; however, we retrieved very few relevant studies from this database. Therefore, we excluded it from our final search. Nevertheless, we are confident that the literature included in our analysis is sufficiently comprehensive and unlikely to have omitted other relevant studies on imagery practice.
- Inthe search appendixes it says February 22 and in the text,it says the 23rd,which was it?
Reply: Thank you for your comment. The actual search date was the 22nd, and the incorrect date in the manuscript was due to our oversight. We have now corrected it in the revised version.
- "Studiesexhibiting more than two,but fewer than four areas designated as unclear risk were categorized as having moderate overall risk”Why did they leave it like that?Because of previous studies?
Reply: Thank you for your comment. Yes, our statistical analysis and risk bias assessment process were modeled after a well-known article published in the BMJ. We have also used this statistical method for more than one article. I personally believe this is currently one of the most advanced and sophisticated approaches for conducting meta-analysis. We have great admiration for this article and its authors, and we have learned all of their meta-analysis methods.
Here is the reference: Noetel, M., Sanders, T., Gallardo-Gómez, D., Taylor, P., Del Pozo Cruz, B., Van Den Hoek, D., Smith, J. J., Mahoney, J., Spathis, J., Moresi, M., Pagano, R., Pagano, L., Vasconcellos, R., Arnott, H., Varley, B., Parker, P., Biddle, S., & Lonsdale, C. (2024). Effect of exercise for depression: Systematic review and network meta-analysis of randomised controlled trials. BMJ, e075847. https://doi.org/10.1136/bmj-2023-075847
- Iwould like you to follow the same line of citing everything in the methodology,in this case,add a citation in the "GRADE"evaluation sect
Reply: Thank you for your comments. I have added the corresponding references in the relevant section.
- "Toevaluate the convergence of the Markov chains,we relied solely on the Rhat 214 Rhat assesses the ratio of between-chain to within-chain variance,where values 215 close to 1 suggest satisfactory convergence.This metric provides a reliable assessment of 216 convergence,ensuring that chains with Rhat values near 1 have thoroughly explored the 217 posterior distribution,thereby enhancing the reliability of model inference."Likewise,to maintain the line of the work,I would recommend you cite some text.
Reply: Thank you for your comments. I have added the corresponding references in the relevant section.
Results
- Theymust detail in their information search that they conducted a grey literature search through citations,in addition to indicating this in the results and
Reply: Thank you for your valuable suggestion. First, we would like to clarify that we did not include any grey literature in our study. All the studies included in our analysis are formally published randomized controlled trials (RCTs). We conducted citation tracking as an additional search strategy and identified 9 additional eligible studies. However, these studies are not grey literature, but peer-reviewed RCTs that met our inclusion criteria. They were missed in the initial screening process due to limitations in our original search strategy.
Additionally, we would like to clarify that the 9 studies identified through citation tracking are no different from the 77 studies identified through the standard screening process in terms of study design, publication status, and eligibility criteria. All are formally published randomized controlled trials. Therefore, we believe that no further clarification or separation of these studies is necessary in the manuscript. We sincerely appreciate your thoughtful suggestion.
- Verygood results section,I have nothing to suggest,it was a good idea to include the supplementary
I insist,you have a perfect command of statistics.I am too far behind to make any useful recommendations.Congratulations.
Reply: Thank you very much for your kind and encouraging comments on our results section. Your recognition is a great source of motivation for our team and provides meaningful affirmation of the long-standing efforts we have invested in this study. We sincerely appreciate your positive feedback.
Discussion
The "Summary of Findings"section was a great way to summarize everything for the reader after such a dense section of results.
- I have thesame question as in the introductioWhether the citation is correct as you say,or whether it's McNeil et al.(2021) … …
Reply: Thank you for the suggestion again. I have already made the revision.
- Iwould like them to end by explicitly stating whether their hypotheses were correct or not (it's understandable when reading the text,but that's the idea of hypotheses)
Reply: Thank you for your valuable comment. In response to your suggestion, we have revised the research hypotheses and clearly stated them in the Introduction section. The two hypotheses are: (1) imagery practice can enhance athletic performance, and (2) longer durations of imagery practice are associated with greater effects.
In the Conclusion section, we explicitly state that the results of our study align with these hypotheses, as the data supported both the positive impact of imagery practice on athletic performance and the trend that longer practice duration tend to produce greater effects. This explicit statement provides clarity on the consistency between our hypotheses and the research findings.( The findings of this study confirm our hypotheses, indicating that imagery practice enhances athletic performance, and that longer practice durations result in more pronounced effects.)
- Thisis a very good discussion section,very thorough in how everything was addressed and justified. I think I would have liked to further emphasize the importance of even the smallest improvement in a high-performance athlete;it may be synonymous with a gold medal,not a silver,but that's a matter of taste,and the work is yours.
Reply: Thank you very much for this suggestion. However, we have decided not to add it. The revisions we’ve made already are extensive, and adding more content may lead to structural issues in the article.
Very good section on limitations.I liked that they stated the reliability of the findings.It demonstrates a great deal of sincerity and objectivity.
Reply: Thank you so much!
Conclusion
Everything is fine.
- As a question,why don't you mentionthe women thing in the conclusion?Wasn't it conclusive enough?I thought it was a very
Reply: Thank you very much for this interesting question. Yes, the current evidence is too limited, and so far, no scholars have specifically studied whether the effects of imagery practice are influenced by gender differences. Therefore, we have chosen to present the main findings in the conclusion, while excluding less robust results from it.
Round 2
Reviewer 2 Report
Comments and Suggestions for Authors
Thank you for the answers and revision, I suggest the article to be accepted.